

# The effect of hair removal and filtering on melanoma detection: a comparative deep learning study with AlexNet CNN

Angélica Quishpe-Usca[1,*], Stefany Cuenca-Dominguez[1], Araceli Arias-Viñansaca[1], Karen Bosmediano-Angos[1], Fernando Villalba-Meneses[1], Lenin Ramírez-Cando[1], Andrés Tirado-Espín[2], Carolina Cadena-Morejón[2], Diego Almeida-Galárraga[1] and Cesar Guevara[3,*]

[1] School of Biological Sciences and Engineering, Yachay Tech University, San Miguel de Urcuquí, Imbabura, Ecuador
[2] School of Mathematical and Computational Sciences, Yachay Tech University, San Miguel de Urcuquí, Imbabura, Ecuador
[3] Quantitative Methods Department, CUNEF Universidad, Madrid, Madrid, Spain
[*] These authors contributed equally to this work.

Corresponding author
Cesar Guevara,
cesar.guevara@cunef.edu

## ABSTRACT

Melanoma is the most aggressive and prevalent form of skin cancer globally, with a higher incidence in men and individuals with fair skin. Early detection of melanoma is essential for the successful treatment and prevention of metastasis. In this context, deep learning methods, distinguished by their ability to perform automated and detailed analysis, extracting melanoma-specific features, have emerged. These approaches excel in performing large-scale analysis, optimizing time, and providing accurate diagnoses, contributing to timely treatments compared to conventional diagnostic methods. The present study offers a methodology to assess the effectiveness of an AlexNet-based convolutional neural network (CNN) in identifying early-stage melanomas. The model is trained on a balanced dataset of 10,605 dermoscopic images, and on modified datasets where hair, a potential obstructive factor, was detected and removed allowing for an assessment of how hair removal affects the model's overall performance. To perform hair removal, we propose a morphological algorithm combined with different filtering techniques for comparison: Fourier, Wavelet, average blur, and low-pass filters. The model is evaluated through 10-fold cross-validation and the metrics of accuracy, recall, precision, and the F1 score. The results demonstrate that the proposed model performs the best for the dataset where we implemented both a Wavelet filter and hair removal algorithm. It has an accuracy of 91.30%, a recall of 87%, a precision of 95.19%, and an F1 score of 90.91%.

## INTRODUCTION

Skin cancers are the most commonly diagnosed group worldwide (*Apalla et al., 2017*). They are generally classified into two major categories: melanoma and nonmelanoma skin cancer. Malignant melanomas may rarely occur in the mouth, intestine, or eye but

more commonly in the skin. Environmental factors as well as genetics play a role in the development of melanoma. Exposure to ultraviolet (UV) radiation from the sun and tanning lamps is taught to be the leading cause of melanoma (*Garibyan & Fisher, 2010*; *Sun et al., 2020*). Although it can affect people of all skin types, melanoma is notably more prevalent among males and individuals with fair skin complexion (*Howlader et al., 2014*). The highest incidence of melanoma is found in Australia, New Zealand, and North America, but it is also becoming increasingly prevalent in Africa, Asia, and South America (*Nikolaou & Stratigos, 2014*). According to the International Agency for Research on Cancer (IARC), melanoma is the most aggressive and prevalent type of skin cancer in the world, with an estimated 325 000 new cases identified worldwide and 57 000 deaths in 2020. If this tendency persists, it is projected that by 2,040, new cases will increase by more than 50% and deaths by 68% (*Arnold et al., 2022*).

Melanoma is considered the most dangerous type of skin cancer because it grows quickly and can spread to any organ. If not treated early, it might result in death or other severe, incurable conditions. Also, visible areas of the skin might be affected, resulting in psychological distress, depression, and emotional turmoil (*Dinnes et al., 2018*; *Vojvodic et al., 2019*). Early detection is crucial for successful melanoma treatment (*Xavier et al., 2016*). Visual examination by medical professionals is typically the initial technique employed in melanoma detection. It involves a comprehensive inspection of abnormal skin lesions, to categorize them as either malignant (melanoma) or benign, based on the ABCDE criteria (asymmetry, border, color, diameter, and evolving) (*Johr, 2002*; *Aljanabi et al., 2020*). It is important to note that the evidence to support the visual inspection's accuracy is insufficiently reported. Various authors agree that its accuracy might be influenced by factors such as the experience of physicians and the diverse appearance of the lesions (*Schwartz et al., 2002*; *See, 2012*; *Dinnes et al., 2018*; *Bansal, Garg & Soni, 2022*). Moreover, it requires significant investments of time, personal, and financial resources, making it a labor-intensive, time-consuming, and error-prone process (*Vestergaard et al., 2008*; *Yu et al., 2016*; *Dabeer, Khan & Islam, 2019*).

Deep-learning methods have emerged as a noninvasive technique for melanoma detection offering a promising approach to overcome the limitations associated with visual inspection. Nowadays, a great amount of work in the detection of skin cancer is concentrated on applying machine-learning techniques to dermoscopic images to improve diagnostic accuracy (*Dildar et al., 2021*). This technique allows efficient automation of skin lesion assessment based on the creation of statistical algorithms for learning purposes (*Char, Shah & Magnus, 2018*; *Fu'adah et al., 2020*). Deep learning can be conceptualized as a hierarchical process for eliciting features, inspired by the workings of the human brain (*Winkler et al., 2019*). These strategies analyze unstructured patterns in dermoscopic images by extracting and organizing deep and precise features that are not easily observed by simple visual inspection. These algorithms have the ability to evaluate large data sets at the pixel level in order to extract melanoma-specific features (*Hagerty et al., 2019*). The usefulness of the method lies in carefully examining every detail of the images and accurately associating it with melanoma, learning to classify by patterns and distinctive features (*Winkler et al., 2019*). This approach not only helps to reduce errors but also

allows a greater amount of data to be analyzed in less time, providing a timely diagnosis and treatment.

Skin lesion classification has recently been addressed in studies using Transfer Learning and Vision Transformers (ViTs) (*Dosovitskiy et al., 2020*; *Xin et al., 2022*). Transformers are a deep learning architecture based on multi-head self-attention mechanisms, originally proposed for sequence-to-sequence tasks in natural language processing (NLP), but now adapted for computer vision tasks such as image classification (*Usman, Zia & Tariq, 2022*). These transformers, known as vision transformers, have recently been applied to classification tasks of biological and epidemiological images of skin lesions. The performance of ViTs for multiclass classification of skin lesions has been compared with convolutional neural networks (CNNs), showing that when trained from scratch, they achieve lower performance (*Maurício, Domingues & Bernardino, 2023*). *Cirrincione et al. (2023)* have also proposed integrated models that combine transformer encoders with masked labels and visual features of CNNs for melanoma classification in ISIC datasets. However, CNNs exhibit higher computational efficiency on standard hardware and have been more researched in the medical imaging domain. In addition, other approaches for feature extraction in image classification studies include local binary pattern (LBP), Data-Efficient Image Transformers (DeiTs), and Convolutional Vision Transformers (CvT) (*Chen et al., 2021*).

CNNs are widely utilized in the field of deep learning for melanoma detection. The underlying principle behind CNNs lies in their architecture, which employs three major types of layers to teach the network how to identify and categorize patterns in images (*Yamashita et al., 2018*). Several convolutional layers are responsible for detecting features in the input image, such as edges and textures. They are usually followed by pooling layers, which aim to shrink large-size feature maps to create smaller feature maps to recognize some general patterns that are only perceptible in resized images (*Nasr-Esfahani et al., 2016*; *Alzubaidi et al., 2021*; *Zhang, 2021*). The third layer, named the fully connected layer, maps the extracted features into a final output, which is the classification. In comparison with traditional machine learning algorithms, CNNs are specifically designed to handle recognition tasks in image spatial structure. One of the main advantages of CNNs is their computational efficiency. Its weight-sharing feature reduces the number of trainable network parameters simplifying the training process and avoiding overfitting.

The VGG16 architecture, created by the Visual Geometry Group at Oxford University, marks a significant evolution in CNNs. Introduced in 2013, this model is renowned for its depth, consisting of 16 layers that include 13 convolutional layers (*Ibrahim et al., 2023*). A key innovation of VGG16 is its use of small $3\times3$ convolutional filters, a departure from larger filters used in earlier models. This design choice enhances its ability to process spatial features in images, aiding in more accurate feature extraction. VGG16's architecture is elegantly simple yet highly effective, making it a popular choice for various applications, especially in image classification tasks like skin cancer detection. Its structure involves multiple layers of convolution, each followed by non-linear activation functions, which contribute to the model's efficiency in learning complex patterns (*Hasan et al., 2021*). For skin cancer classification, VGG16 is often adapted through transfer learning, where

its pre-trained layers are fine-tuned to identify specific features of skin lesions. This methodology leverages the network's in-depth learning from vast datasets to provide precise classifications between malignant and benign skin conditions.

The reported variations in performance and accuracy across different segmentation tasks and datasets can be attributed to the design variations among different architectures (*Perez, Avila & Valle, 2019*). Numerous studies have shown that CNNs are highly effective for classification, localization, detection, segmentation, and registration in dermoscopic image analysis. These investigations have led to significant advancements in the development and refinement of CNN-based algorithms specifically designed to accurately and efficiently detect melanoma.

Table 1 summarizes previous studies on CNN-based skin cancer classification, providing insights into their efficiency, effectiveness, strengths, and weaknesses.

In this context, studies have evaluated strategies to measure model performance, highlighting the accuracy (ACC) metric emerges as a global metric that provides a comprehensive measure of classifier accuracy. Balanced accuracy (BACC) is the average between the sensitivity and the specificity and is prioritized for unbiased evaluation (*Brodersen et al., 2010*). Specificity (SP) for negative cases and sensitivity (SE) for positive cases are analyzed. Precision (PR) excels in the accurate identification of positive cases. These criteria are meticulously chosen to provide a comprehensive and fair assessment of the models' ability to classify efficiently, offering a complete picture of their performance.

Segmentation of dermoscopic images using CNNs can be significantly influenced by various factors, such as low contrast between the lesion and healthy skin, color variations, the presence of air bubbles, and the obstruction caused by hair covering the lesions. Hairs and their shadows on the skin may occlude relevant information and can cause segmentation algorithms to commit errors in estimating texture measures (*Oliveira et al., 2016*). Consequently, hair removal might be considered a crucial preprocessing step in training CNNs for accurate segmentation.

Traditional approaches have treated hair removal as a hair detection and occlusion inpainting problem, employing filter-based methods (*Nguyen, Lee & Atkins, 2010*; *Eltayef, Li & Liu, 2017*; *Li et al., 2021*). Recent advancements utilize multi-scale curvilinear matched filters for hair detection, followed by region-growing algorithms and linear discriminant analysis for precise localization. Modern data-driven approaches employ deep neural network auto-encoders trained on finely annotated datasets (*Bardou et al., 2022*). However, constructing paired hair-containing and hair-removed images for training these algorithms remains challenging due to their high cost and limited availability. While the effect of hair in dermoscopic images on CNN performance has been studied, it has yet not been sufficiently reported. Some studies have highlighted the significant improvement in classification performance achieved by removing hair from dermoscopic images (*Kim & Hong, 2021*; *Li et al., 2021*). Various types of filters such as Gaussian, Average, Median and Wiener have also been used to reduce noise and smooth dermoscopic images of medical images combined with machine learning algorithms (*Bektaş et al., 2018*).

The main objective of this study is to assess the impact of hair removal combined with different filtering techniques on the performance of a CNN utilizing the AlexNet

**Table 1 References of skin cancer segmentation with typical CNN frameworks in the literature.** Only the metrics for the best-performing architectures or datasets are presented in studies that involve the comparison of multiple architectures or datasets.

| Ref/Year | Dataset | Architecture | Highlights | Limitations | Performance |
|---|---|---|---|---|---|
| (*Ameri, 2020*)/2020 | HAM10000 | AlexNet | The suggested approach eliminates the need for complex procedures of lesion segmentation and feature extraction by taking an unprocessed image as input and learning directly from the raw data. | Only 3,400 images from the dataset were used due to the need for an equal number of benign and malignant images for training. | ACC = 0.84, SE = 0.81, SP = 0.88 |
| (*Yao et al., 2022*)/2022 | ISIC-2017, ISIC-2018, ISIC-2019, 7-PT | RegNetY-3.2GF | The study proposes a novel Multi-weighted New Loss method to address the issue of class imbalance and improve accuracy in detecting key classes such as melanoma. RegNetY performed the best on the ISIC2018 dataset. | Almost all publicly accessible skin disease image datasets suffer a problem of severe data imbalance that might affect the performance of CNNs. | BACC = 0.858 |
| (*Perez, Avila & Valle, 2019*)/2019 | ISIC-2017 | Inception-ResNet-v2, MobileNetV2 , PNAS-Net, ResNet , SENet, Xception, VGG16, VGG19, and DenseNet | The authors systematically assessed the factors that impact the selection of a CNN structure by examining 13 different factors across nine models. | The article's dataset has limitations as it is smaller in size compared to other studies and its exclusive focus on classifying melanoma. | Top-1 ACC = 0.827 |
| (*Javid et al., 2023*)/2023 | A recompilation of ISIC datasets | ResNet 50, EfficientNet B6, Inception V3, and Xception | The results obtained from each individual model are inputted into a meta-learner in order to combine and utilize the outputs from these models to make a final prediction. | – | ACC = 0.935, SE = 0.9, PR = 0.96, F1 score = 0.92 |
| (*Alwakid et al., 2022*)/2022 | HAM10000 | Modified version of Resnet-50 | The proposed method suggests utilizing DL to precisely extract a lesion zone. The approach involves enhancing the image quality using ESRGAN and then using segmentation to isolate Regions of Interest (ROI). | To showcase the effectiveness of the proposed technique, it is necessary to conduct more experiments on a sizable and intricate dataset that encompasses potential cancer cases. | ACC = 0.86, SE = 0.86, PR = 0.84, F1 score = 0.86 |

**Table 1** (*continued*)

| Ref/Year | Dataset | Architecture | Highlights | Limitations | Performance |
|---|---|---|---|---|---|
| (*Raza et al., 2022*)/2022 | Clinical repositories in Korea | InceptionV3, Xception, InceptionResnetV2, DenseNet121, VGG16 | A novel stacked ensemble framework has been introduced in the study, specifically designed to augment generalizability and bolster robustness in the context of acral lentiginous melanoma classification. | – | ACC = 0.979, SE = 0.978, PR = 0.98, F1 score = 0.98 |
| (*Gupta & Mesram, 2022*)/2022 | ISIC 2016–17 | AlexNet, DenseNet-121 | The study suggests a mixed CNN model that involves merging a pre-trained AlexNet CNN model with an optimized pre-trained DenseNet-121 CNN model. | Numerous healthcare institutions possess substantial patient data; however, they face challenges in making this information accessible to the public due to privacy concerns. | ACC = 0.9065, SE = 0.91, PR = 0.9065, F1 score = 0.91 |
| (*Esteva et al., 2017*)/2017 | Clinical repositories | GoogleNet, Inception v3 | The model is adapted to be used on mobile devices. It is predicted that by the year 2021, there will be approximately 6.3 billion smartphone subscriptions globally. | Further investigations are necessary to evaluate how this method performs in a clinical setting. | ACC = 0.721 |

**Notes.**

*Alwakid et al. (2022)*
*Ameri (2020)*
*Esteva et al. (2017)*
*Gupta & Mesram (2022)*
*Javid et al. (2023)*
*Perez, Avila & Valle (2019)*
*Raza et al. (2022)*
*Yao et al. (2022)*

architecture for early melanoma detection. We use a dataset that is the collection of dermoscopic images from multiple freely available International Skin Imaging Collaboration (ISIC) and HAM 10,000 datasets. Additionally, modified datasets were obtained by detecting and subsequently removing hair by a morphological algorithm. Different filtering techniques are used to enhance image quality and remove noise and are applied before hair removal for further comparison. The main contributions of this research are: (i) combining a hair removal algorithm with four different filtering techniques (Fourier, Wavelet, Average blur, and Low-pass) ; (ii) conducting a comparison, in terms of accuracy, recall, precision, and F1 score, for each filter combination to determine its effect on the CNN performance for melanoma classification ; (iii) using a balanced dataset comprising dermoscopic images from various multi-class skin lesion datasets. However, a notable limitation is the similarity in metric results, prompting the need for further validation through one-factor analysis of variance.

## Related work

In recent research, significant progress has been observed in the use of CNNs and other artificial intelligence techniques for skin cancer classification. These approaches have shown great potential to help specialists reduce the time and resources required for diagnosis, which, in turn, allows for more appropriate treatment. A study by *Leiter, Keim & Garbe (2020)* compared expert opinion with artificial neural networks, and found that the software established a sensitivity of 95% and specificity of 88%, results comparable to those reported by dermatologists.

Several researchers have proposed specific approaches for skin cancer classification using CNN and other deep-learning techniques. Among these approaches, *Yanchatuña et al. (2021)* used a combination of CNNs and support vector machines (SVMs) to detect and classify skin cancer, obtaining an average accuracy between 80.67% and 90%, with an outstanding performance of 90.34% for the AlexNet plus SVM model. *Popescu, El-Khatib & Ichim (2022)* developed a skin lesion classification system involving multiple CNNs trained on the HAM10000 dataset, capable of predicting seven skin lesions, including melanoma. *Ameri (2020)* presented a deep CNN using transfer learning with AlexNet, eliminating the need for complex segmentation and feature extraction procedures by automatically learning useful features from raw images.

Other researchers explored the use of pre-trained networks (such as AlexNet and VGG16) in learning transfer and as feature extractors (*Gulati & Bhogal, 2019*). VGG16 with transfer learning was found to outperform other techniques in terms of accuracy and efficiency. This is attributed to its deep network structure and the ability to adapt to specific tasks through the use of pre-trained weights, resulting in better performance in the classification of skin lesions, as has been documented in various investigations (*Anand et al., 2022*). *Alwakid et al. (2022)* proposed an approach that incorporates ESRGAN, segmentation techniques, and a CNN along with a modified version of Resnet-50 for accurate classification. Similarly, hybrid models were presented that combine CNNs with SVM classifiers (*Keerthana et al., 2023*) or dimension reduction techniques such as PCA (*Olayah et al., 2023*). *Naeem et al. (2022)* developed the SCDNet model, which combines

VGG16 with CNN and compares its accuracy with pre-trained classifiers in the medical domain.

In addition, comparisons of various CNN architectures were performed, where GoogleNet proved to be the most accurate (74.91% and 76.08%) in both training and test sets (*Aljohani & Turki, 2022*). *Acosta et al. (2021)* proposed a two-stage approach using CNN-based masks and regions and a ResNet152 structure to classify lesions as "benign" or "malignant". *Nida et al. (2019)* introduced an automated melanoma region segmentation method based on a deep region-based convolutional neural network (RCNN) that accurately detects multiple affected regions. *Gouda & Amudha (2020)* presented LCNet, a model that requires no preprocessing or computation of specific features. *Tahir et al. (2023)* compared their novel DSCC_Net model with other deep benchmark networks, demonstrating improved accuracy.

Finally, *Bansal, Garg & Soni (2022)* introduced methods to remove hair in dermatoscopic images and used integrated features extracted using manual techniques and deep learning models to improve melanoma detection. Experimental results highlighted improvements in accuracy compared to approaches that do not apply preprocessing or use manual or deep learning features separately.

# MATERIALS & METHODS

This study proposes a method for training and testing a CNN with AlexNet architecture for early melanoma detection. Four filtering techniques are implemented before a morphological hair removal algorithm for further comparison. First, dermoscopic images of skin lesions are obtained and preprocessed from the so-called "MSCD10000 dataset" (*Javid et al., 2023*). Then, the different layers of the CNN architecture are designed and finally, the model performance is evaluated using 10-fold cross-validation and four metrics. The software used in the current study is Python.

## MSCD10000 dataset

MSCD10000 dataset stands for Melanoma Skin Cancer Dataset of 10,000 Images and consists of 10,605 dermoscopic images scaled to 300x300 pixels (*Javid et al., 2023*). These images were collected and resized from various publicly available ISIC (ISIC 2019, ISIC 2018) and HAM 10,000 datasets to create a balanced dataset with both benign and malignant classes. The dataset was downloaded from Kaggle at the following link: https://www.kaggle.com/datasets/hasnainjaved/melanoma-skin-cancer-dataset-of-10000-images. It is organized into two main categories, *train* and *test*. Within each category, the dataset is further divided into *benign* and *malignant* classes. Table 2 provides details on the distribution of dermoscopic images across these categories.

## Hair removal and filtering

Before the hair removal process on dermoscopic images, we used four different filtering techniques to reduce noise and improve image quality: Fourier, Wavelet, Average blur, and Low-pass. Each filter was integrated using *Python 3.x*, *cv2* libraries, and *OpenCV*, with our codebase adeptly adjusted to accommodate the unique requirements of each filter type. To

**Table 2 Dataset distribution.**

| | Train | | Test | | Total (%) |
| --- | --- | --- | --- | --- | --- |
| | Number | Percentage (%) | Number | Percentage (%) | |
| Malignant | 4,605 | 43.42 | 500 | 4.71 | 5,105 (48.14%) |
| Benign | 5,000 | 47.14 | 500 | 4.71 | 5,500 (51.86%) |
| Total | 9,605 | 90.57 | 1 000 | 9.43 | |

illustrate, when implementing the low-pass filter, we applied a Gaussian blur with a 3×3 kernel size. *PyWavelets* for the implementation of Wavelet filters was used. The choice of these libraries allowed for efficient and reproducible processing of the images.

The process of filter implementation and hair removal is illustrated in Fig. 1. First, the images were converted to grayscale. They were then processed with a black top-hat filter and thresholding, followed by an inpainting method based on cv2.INPAINT_TELEA to restore the image (*Telea, 2004*). This process was performed using a 17x17 size core for the MORPH BLACKHAT operation, highlighting dark spots, which correspond mainly to hair, in the dermoscopic images (*Ashraf et al., 2022*).

After applying the filtering techniques, the grayscale images were subjected to a morphological operation for hair detection. We use a threshold value of 10 for binarization of the images, classifying them into pixels that represent hair (greater than 10) and background pixels (less than or equal to 10). The images were stored in separate folders labeled "with hair" and "without hair."

The computational complexity of the hair removal process was considerable. On average, processing each image folder took between 2:30 and 3 h, with a memory requirement of approximately 6 GB. This time includes the application of the filters, the grayscale conversion, the morphological operation, the thresholding, and the filling method (inpainting). These factors were essential to ensure the viability of the method in both research and clinical practice.

## Image pre-processing

The recommended optimal size for input images to ensure compatibility with the CNN architecture's input layer is $224 \times 224 \times 3$ (*Krizhevsky, Sutskever & Hinton, 2012*). The resized images are converted into a PyTorch tensor and then, the tensor values for each image are normalized sequentially with the mean and standard deviation values for each color channel. That is, scaling the values so that they have a mean of 0 and a standard deviation of 1 (*LeCun et al., 1998*; *Glorot & Bengio, 2010*; *Ioffe & Szegedy, 2015*).

Furthermore, data augmentation is used to address overfitting issues and improve the model's capacity to generalize to new data by increasing the number of training samples for each class through random transformations not only for imbalanced datasets but also for datasets of any size (*Perez et al., 2018*; *Ali et al., 2022*). We applied different transformations, which included random horizontal flipping with a 50% probability, random rotations of up to 10 degrees, random adjustments in brightness, contrast, saturation, and hue within a maximum range of 0.2, as well as color channel normalization and conversion to the torch. Tensor format to effectively utilize the images as inputs.

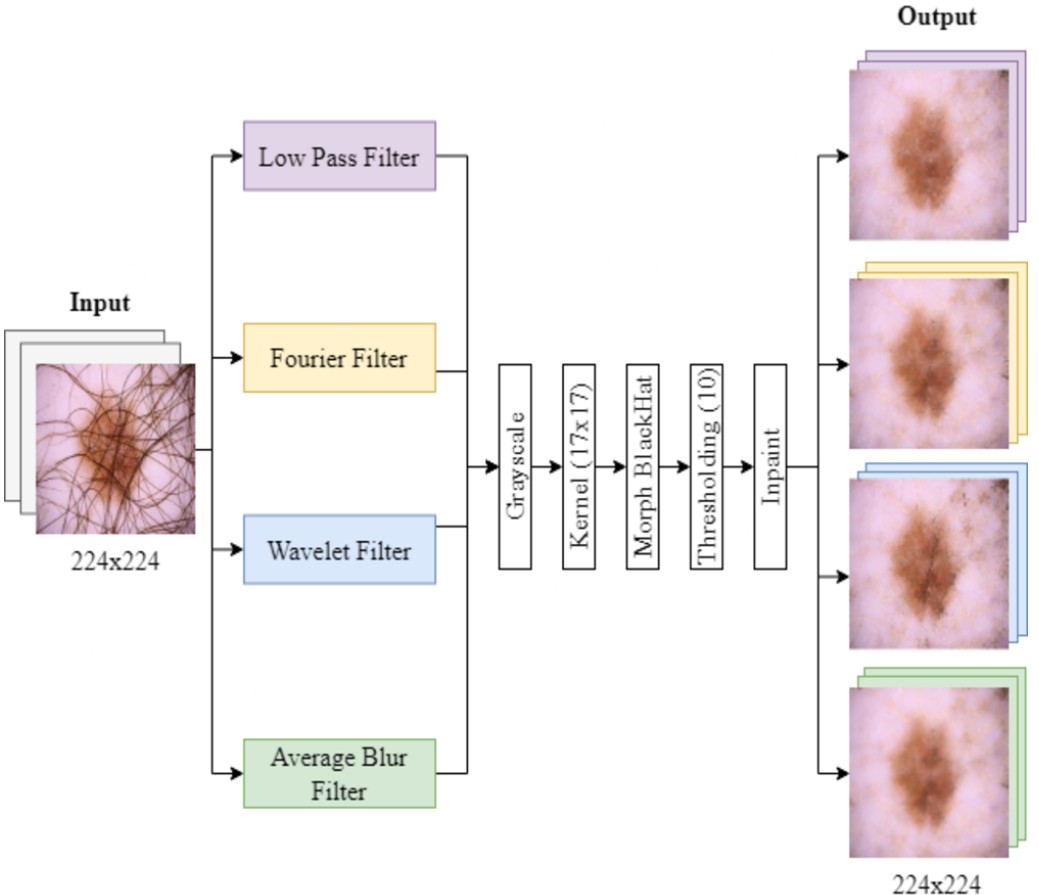

**Figure 1  Filtering and hair removal algorithm.** Images are first turned grayscale, then processed with a black top-hat filter and thresholding, and lastly, an inpainting method restores the image.

## AlexNet architecture

A visual representation of the AlexNet architecture used in this study is provided in Fig. 2. It was trained with hyperparameters carefully chosen to balance effective model convergence and computational efficiency. The learning rate was set to 0.001 to control parameter updates, while a momentum of 0.9 accelerated convergence by incorporating past gradients. A weight decay of 0.001 added regularization to prevent overfitting. The Cross-Entropy Loss function, suitable for multi-class or binary classification, was employed to measure prediction accuracy. A batch size of 64 determined the number of samples processed in each iteration, balancing memory constraints and training speed. The model underwent 30 epochs.

The initial layout encompasses five convolutional layers, each using specialized filters to identify specific features, such as contours and textures, present in the original image. These convolutional layers alternate with maximum aggregation layers, designed to condense the feature maps and simplify the recognition of more holistic patterns, particularly those that might remain unnoticed in larger images. The first, second, third, and fifth convolutional

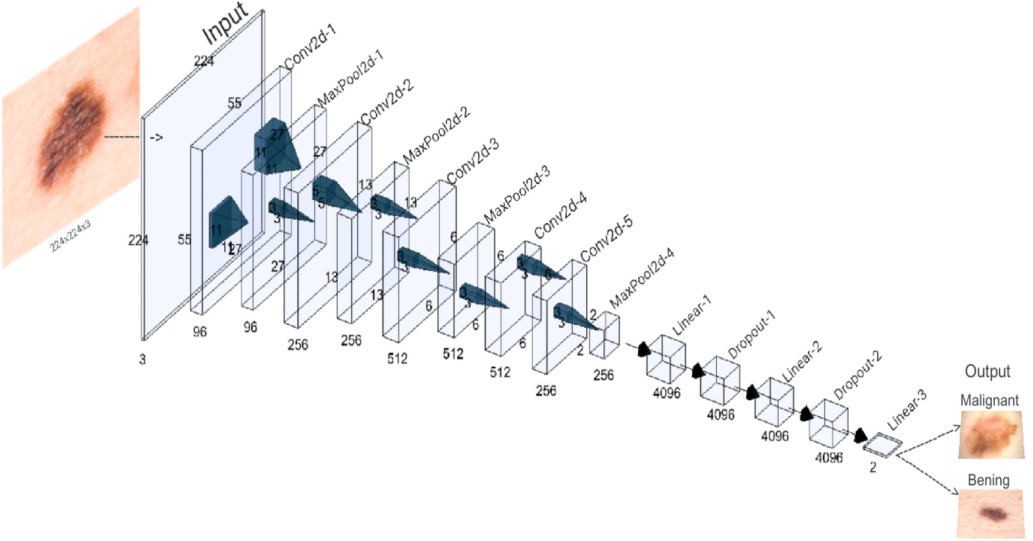

**Figure 2** **AlexNet architecture of the proposed CNN.** A new adaptive layer is added between the convolutional and fully connected layers to modulate the binary output size.

layers are seconded by maximum aggregation layers. An additional classifier module, consisting of three fully connected layers, is introduced to map the extracted features and thus generate the final binary classification.

In a variation on the original AlexNet structure, a new adaptive layer is incorporated between the convolutional and fully connected layers. This dynamically adjustable layer modulates the size of the output tensor to fit a standardized 6x6 dimension. This adjustment leads to an optimization of the model's performance, allowing it to efficiently adapt to the diversity of features present in the medical images under analysis.

## Performance evaluation

We employed the K-fold cross-validation technique for evaluating the average performance of the model in terms of the average training and test loss. The *train* directory of each dataset, original and modified, is divided into 10 folds of equal size. The choice of the value for *k* was determined based on the dataset's size. The model undergoes 10 training iterations, each consisting of 30 epochs, with one fold serving as the validation set in each iteration and the remaining 9 folds used for training.

In addition, a quantitative evaluation of the CNN performance was carried out using four common metrics: accuracy, sensitivity, precision, and F1 score. True negatives (TN) and true positives (TP) refer to the accurate classification of negative and positive instances, respectively. On the other hand, false negatives (FN) and false positives (FP) refer to the incorrect classification of positive and negative instances, respectively.

$$Accuracy = \frac{True\ detected\ melanoma\ cases\ (TP + TN)}{All\ cases}$$

$$Recall = \frac{True\ detected\ melanoma\ cases\ (TP)}{All\ melanoma\ cases\ (TP + FN)}$$

$$Precision = \frac{True\ detected\ melanoma\ cases\ (TP)}{True\ and\ false\ detected\ melanoma\ cases\ (TP + FP)}$$

$$F1\ score = \frac{Precision * Recall}{Precision + Recall}.$$

## RESULTS AND DISCUSSION

In this section, we describe the effectiveness of the proposed CNN for detecting melanoma. We first evaluate the impact of hair removal on its performance through a comparison of the results for the model trained in the original and the modified datasets, where filtering techniques and hair removal were applied. Subsequently, we compare the outcomes of the four metrics obtained from the model trained on the best-performing dataset to those achieved by existing studies in the field.

### Hair removal results

To ensure optimal model selection, we performed model evaluation every 30 epochs for each fold. Figure 3 shows the training and testing (validation) loss *vs.* epochs for the different datasets (original and modified) presenting a detailed analysis of the evolution of the loss throughout the training. The model curves are not smooth, instead, a lot of fluctuations are observed. These fluctuations could be attributed to several factors, with batch size being one of them. Notably, prior research suggests that a batch size of 32 often leads to optimal results in neural network training and as batch size increases, the convergence can be negatively impacted (*Keskar et al., 2017*). However, it is observed that the training loss is reduced to 0.1 for all of the model loss curves, which can be considered a good sign for correct segmentation results as other studies conclude (*Ishida et al., 2021*; *Guefrechi et al., 2021*).

The loss curve for the original images shows an increase in both training and testing loss at 30 epochs, which is not observed in the rest of the curves. Furthermore, even when data augmentation was used, some overfitting was observed when using the Fourier filter before hair removal, which can cause a decrease in generalization ability.

The overall CNN performance is determined by comparing the results of the four metrics between the original dataset and the modified datasets. The results in Table 3 reveal that the model performs the best among three of the four metrics with images where we have applied the Wavelet filter for noise reduction combined with hair removal. Here, the model achieves an accuracy of 91.30%, and a recall of 87%, while the low-pass filter yields a slightly higher recall of 89.80%. Additionally, when utilizing the Wavelet filter, the model has a precision of 95.19% and an F1 score of 90.91%.

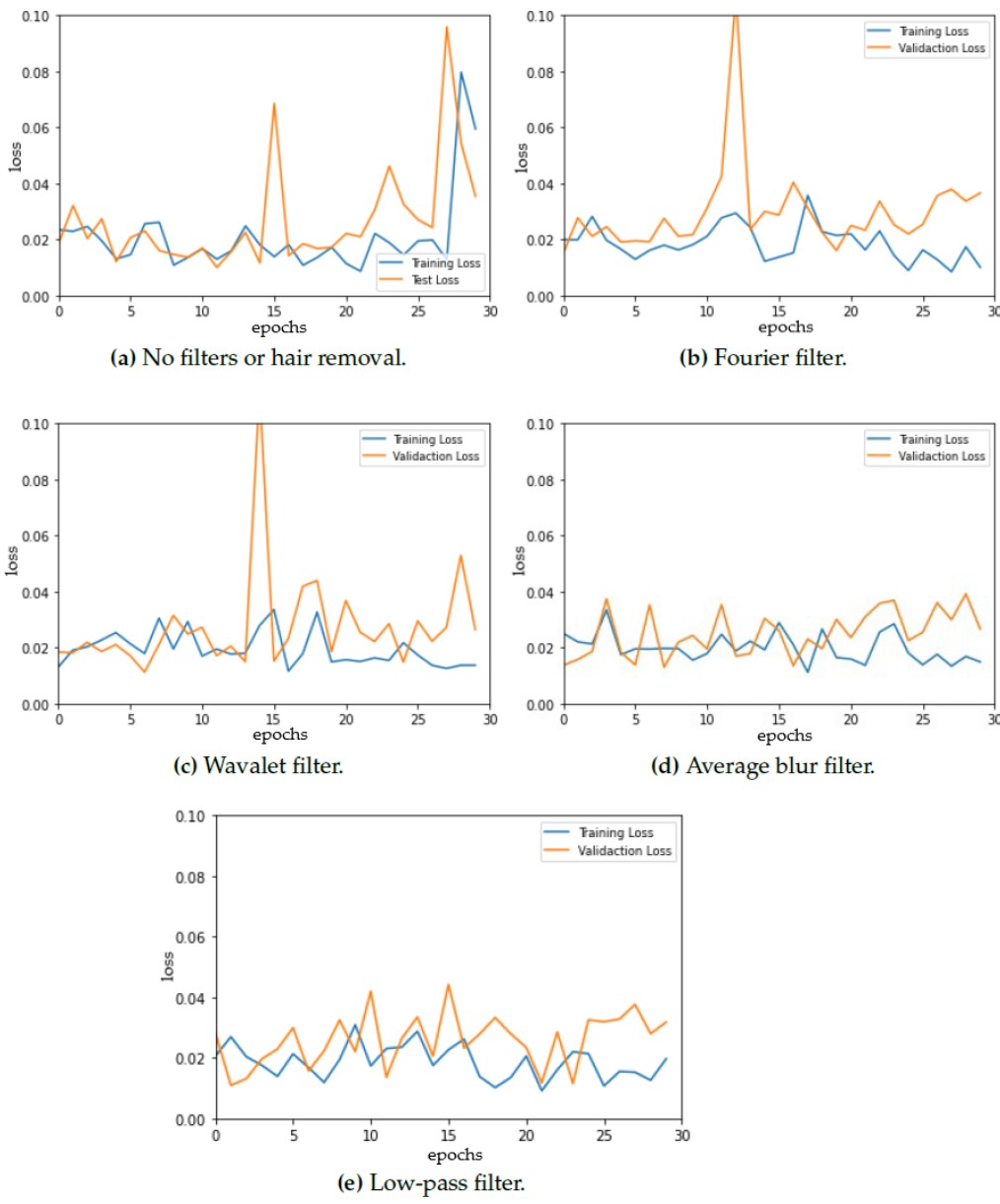

**Figure 3** Model loss for (A) original dermoscopic images *vs.* modified images that combine (B) Fourier filter, (C) Wavelet filter, (D) average blur, and (E) low-pass filter with hair removal.

The results indicate that hair removal improves the model's accuracy and precision when it is combined with the Wavelet filter allowing the model to not only make correct predictions overall but also minimize false positives. Wavelet transform has already been used to equalize the noise created by fine hairs in dermoscopic images as well as to increase the algorithm's detection speed in hybrid models combining deep learning and machine learning (*Suiçmez et al., 2023*). Some of the highest accuracy and sensitivity scores in the literature have been obtained when using Wavelet filters (*Narasimhan & Elamaran, 2016*).

**Table 3** Trained model results on the original images from the dataset and the images where different filtering techniques were applied and hair was removed.

| Metrics | Filtering technique combined with hair removal | | | | |
|---|---|---|---|---|---|
| | None | Fourier | Wavelet | Average blur | Low-pass |
| Accuracy (AC) | 91.10% | 90.90% | 91.30% | 90.80% | 90.00% |
| Recall (SE) | 87.60% | 88.20% | 87.00% | 87.40% | 89.80% |
| Precision (PR) | 94.19% | 93.23% | 95.19% | 93.78% | 90.16% |
| F1 score (F1 sc) | 90.78% | 90.65% | 90.91% | 90.48% | 89.98% |

On the other hand, the model trained on the original images exhibits the second-highest accuracy and precision of 91.10% and 94.19%, respectively. However, the second highest sensibility is reached when using the Fourier filter combined with hair removal.

These results are supported by the graphical representation of a box plot (Fig. 4), which visually points out the significant difference between the filters as a function of their performance. In Fig. 4, it is important to note that the $Y$-axis of the plot does not represent a specific metric, but rather the combination of four different metrics, thus providing a comprehensive view of each filter's performance. The length of the boxes in the figure indicates the variability in the performance of each filter: those with longer boxes exhibit higher variability, while those with shorter boxes exhibit lower variability. This observation is supported by examining the confidence intervals, which are considerably wider for the Wavelet filter compared to the other filters. However, it is pertinent to note that, despite this superiority, the confidence intervals for the Wavelet filter are also the widest, indicating considerable variability in its performance. This observation raises the possibility that, although the Wavelet filter excels in general terms, its performance may fluctuate significantly in different scenarios, which is why we consider that further investigation into the robustness and generalizability of the filter's performance in diverse settings is needed.

Furthermore, it is important to note that the significance of this improvement in relation to variations in the initialization of network parameters or other potential factors is not extensively discussed since hyper-tuning techniques were not applied to the network used in this study. It could be suggested to continually evaluate and potentially fine-tune these hyperparameters and initialization methods based on the characteristics of the melanoma dataset to ensure robust training and accurate classification results.

Overall, these outcomes strongly suggest the model's competence in effectively distinguishing between benign and malignant skin lesions, as evidenced by its robust performance metrics. CNNs offer significant advantages for physicians in clinical settings since they can process large volumes of data quickly and consistently, continuously learn new information, and offer objective assessments (*Krishnan et al., 2023*). For instance, the study by *Haenssle et al. (2018)* compared a CNN's diagnostic performance with that of a large international group of 58 dermatologists, revealing the CNN's superior performance. The CNN exhibited a higher specificity (82.5%) compared to dermatologists at both level-I (71.3%, $P < 0.01$) and level-II (75.7%, $P < 0.01$) sensitivities of 86.6% and 88.9%, respectively. Nonetheless, experts maintain an edge in handling complex or ambiguous skin lesion cases.

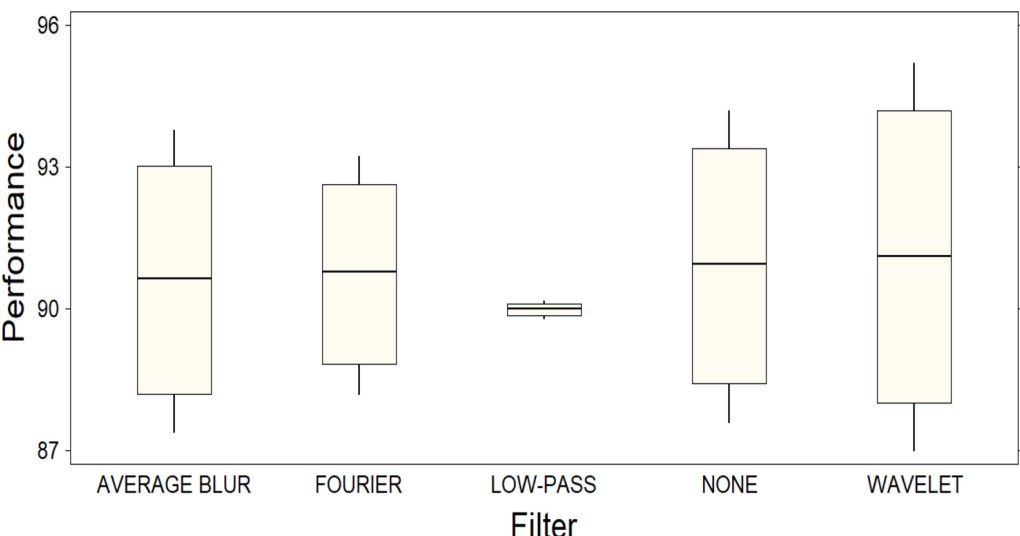

**Figure 4** **Box plot representing the distribution of performance among different filters.** A significant difference between the filters is observed, highlighting the superior performance of the Wavelet filter compared to the others.

**Table 4** **Quantitative comparison of the results for the different metrics.** All values were rounded to 3 decimal places.

| Reference | Model | AC | SE | PR | F sc |
|---|---|---|---|---|---|
| (*Ameri, 2020*) | AlexNet | 0.840 | 0.810 | – | – |
| (*Javid et al., 2023*) | Some | 0.935 | 0.900 | 0.960 | 0.920 |
| (*Alwakid et al., 2022*) | Modified Resnet-50 | 0.860 | 0.860 | 0.840 | 0.860 |
| (*Raza et al., 2022*) | Some | 0.979 | 0.978 | 0.980 | 0.980 |
| (*Gupta & Mesram, 2022*) | Alexnet, DenseNet-121 | 0.907 | 0.910 | 0.906 | 0.910 |
| Proposed 1 | Modified AlexNet | 0.913 | 0.870 | 0.952 | 0.909 |
| Proposed 2 | Modified AlexNet | 0.910 | 0.876 | 0.942 | 0.908 |

**Notes.**
*Alwakid et al. (2022)*
*Ameri (2020)*
*Gupta & Mesram (2022)*
*Javid et al. (2023)*
*Raza et al. (2022)*

## Comparison with related work

We validated our model's performance against two setups: Wavelet filter with hair removal (Proposed 1) and original images without hair removal (Proposed 2). These results are contrasted with some well-established recent studies in the field shown in Table 1. As can be seen in Table 4, the proposed model was optimal in terms of the four metrics: accuracy (AC), recall (SE), precision (PR), and F1 score (Fsc). It has one of the highest accuracy and precision values.

Single-factor analysis of variance (ANOVA) was applied to investigate the existence of statistically significant differences in performance among various neural network models.

**Table 5** Results of the single-factor analysis of variance (ANOVA) based on the data presented in Table 4.

|  | DF* | SS** | MS*** | F**** | P***** |
|---|---|---|---|---|---|
| Models | 6 | 0.91 | 0.15 | 4.55 | 0.0042 |
| Error | 21 | 0.70 | 0.03 | – | – |
| Total | 27 | 1.61 | – | – | – |

**Notes.**
*DF (Degrees of Freedom).
**SS (Sum of Squares).
***MS (Mean Square).
****F (F-statistic).
*****P ( *p*-value).

This method will provide a comprehensive perspective on whether at least one model is statistically different from the others with respect to the performance metrics under evaluation. The results, obtained through Statistix 10 software and presented in the table below, reveal an F-statistic of 4.55 with an associated *p*-value of 0.0042 as shown in Table 5.

The associated *p*-value being below the critical threshold of 0.05, leads to the rejection of the null hypothesis. Consequently, the existence of at least one significant difference in performance between the various neural network models is established.

Overall, the proposed model exhibits competitive performance when compared to the referenced studies. It achieves similar levels of accuracy, recall, precision, and F1 scores, emphasizing its efficacy in detecting melanoma in dermoscopic images. However, it is important to acknowledge that there is still room for improvement to develop and compare new filters for noise removal that might enhance hair removal outcomes in the field of melanoma detection.

## CONCLUSIONS

This study focused on improving the performance of a CNN with AlexNet architecture for early melanoma detection by integrating hair removal and various filtering techniques. The model showed improvements in accuracy and precision when Wavelet filtering combined with hair removal was applied. Thus, filtering and hair removal improve image quality and enhance the model's ability to correctly classify images as melanoma. Finally, the proposed model on images from the original and modified dataset was evaluated and compared with other models in the field, demonstrating high levels of accuracy, precision, recall, and F1 score. In percentage terms, when evaluating the proposed model on the original dataset it achieved an accuracy of 91.10% and 91.30% on the modified images.

Overall, this study contributes to the development of a reliable and accurate system for melanoma detection using CNN. Future research directions could involve enhancing filtering techniques, exploring combinations of different filters, and developing adaptive filters to evaluate their impact on CNN performance concerning image quality and noise reduction. Additionally, the focus will shift towards implementing these systems in medical environments using convolutional networks, with the potential to significantly reduce mortality rates, enhance diagnostic precision, and streamline healthcare processes. The

innovative applications of this technology extend to vital domains such as radiology, digital pathology, and telemedicine, promising notable advancements in medical diagnostics.

### Funding
This work was supported by CUNEF University and Universidad Yachay Tech. The funders had no role in study design, data collection and analysis, decision to publish, or preparation of the manuscript.

### Grant Disclosures
The following grant information was disclosed by the authors:
CUNEF University and Universidad Yachay Tech.

### Competing Interests
The authors declare there are no competing interests.

### Author Contributions
- Angélica Quishpe-Usca conceived and designed the experiments, performed the experiments, analyzed the data, performed the computation work, prepared figures and/or tables, authored or reviewed drafts of the article, and approved the final draft.
- Stefany Cuenca-Dominguez conceived and designed the experiments, performed the experiments, analyzed the data, performed the computation work, prepared figures and/or tables, authored or reviewed drafts of the article, and approved the final draft.
- Araceli Arias-Viñansaca conceived and designed the experiments, performed the experiments, analyzed the data, performed the computation work, prepared figures and/or tables, authored or reviewed drafts of the article, and approved the final draft.
- Karen Bosmediano-Angos conceived and designed the experiments, performed the experiments, analyzed the data, performed the computation work, prepared figures and/or tables, authored or reviewed drafts of the article, and approved the final draft.
- Fernando Villalba-Meneses conceived and designed the experiments, performed the experiments, analyzed the data, performed the computation work, prepared figures and/or tables, authored or reviewed drafts of the article, and approved the final draft.
- Lenin Ramírez-Cando conceived and designed the experiments, performed the experiments, analyzed the data, performed the computation work, prepared figures and/or tables, authored or reviewed drafts of the article, and approved the final draft.
- Andrés Tirado-Espín conceived and designed the experiments, performed the experiments, analyzed the data, performed the computation work, prepared figures and/or tables, authored or reviewed drafts of the article, and approved the final draft.
- Carolina Cadena-Morejón conceived and designed the experiments, performed the experiments, analyzed the data, performed the computation work, prepared figures and/or tables, authored or reviewed drafts of the article, and approved the final draft.

- Diego Almeida-Galárraga conceived and designed the experiments, performed the experiments, analyzed the data, performed the computation work, prepared figures and/or tables, authored or reviewed drafts of the article, and approved the final draft.
- Cesar Guevara conceived and designed the experiments, performed the experiments, analyzed the data, performed the computation work, prepared figures and/or tables, authored or reviewed drafts of the article, and approved the final draft.

### Data Availability

The data is available at Mendeley: Quishpe-Usca, Angélica; Cuenca-Dominguez, Stefany; Arias-Viñansaca, Araceli; Bosmediado-Angos, Karen; Villalba-Meneses, Fernando; Ramírez, Lenin; Tirado-Espín, Andrés; Cadena-Morejón, Carolina; Guevara, Cesar; Almeida-Galárraga, Diego (2024), ''CNN for Melanoma Detection Data'', Mendeley Data, V3, doi: 10.17632/ggh6g39ps2.3.

The original data is available at the Kaggle:

Available at https://www.kaggle.com/datasets/hasnainjaved/melanoma-skin-cancer-dataset-of-10000-images.

### Supplemental Information

Supplemental information for this article can be found online at http://dx.doi.org/10.7717/peerj-cs.1953#supplemental-information.

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
