# Peer review of "The effect of hair removal and filtering on melanoma detection: a comparative deep learning study with AlexNet CNN"

_PeerJ Computer Science, doi:10.7717/peerj-cs.1953_

## Round 0.1 · original submission · Major Revisions

Dear authors,
You are advised to critically respond to all comments point by point when preparing a new version of the manuscript and while preparing for the rebuttal letter. Please address all the comments/suggestions provided by the reviewers.

Reviewer 1 has suggested that you cite specific references. You are welcome to add it/them if you believe they are relevant. However, you are not required to include these citations, and if you do not include them, this will not influence my decision.

Kind regards,
PCoelho

·

Basic reporting

In this work, the authors investigates the use of deep learning for early melanoma detection, focusing on an AlexNet-based CNN model trained on a dataset of 10,605 dermoscopic images. They also examine the impact of hair removal on image analysis, employing a novel morphological algorithm and comparing various filtering techniques, including Fourier, Wavelet, average blur, and low-pass filters. Despite the good structure of the manuscript and the satisfactory use of the English language, some concepts need to be focused on improving paper quality:

Experimental design

- In the introductory section, the contributions are not clear or little specified. Please improve this section of the manuscript by highlighting and listing the “contributions” points of this research and the work's limitations;

- It is also necessary to better explain why a deep-learning approach is needed for the detention of Melanoma in clinical practice. Please highlight this critical aspect from the abstract by comparing it with the actual clinical diagnosis today;

Validity of the findings

- The literature lacks relevant technical approaches used in engineering and computer science that make use of datasets involving biological and epidemiological data and images to classify diseases and/or disorders. Some relevant papers used in the same field but also used in other fields should be cited and discussed in the ‘introduction’ section: i.e., 10. 10.3390/s23125677 (for Melanoma detection and classification), and 10.1109/UBMK.2018.8566380 (in Mammography Images classification). These recommended papers are essential for integrating your methods in different fields and domains, providing a comprehensive view for readers;

- The paper needs a better explanation of the choice of hyper-parameters used for your approach in the simulation results. E.g. threshold, cost function, learning rate, etc. Has there been any hyper-parameter tuning?

Additional comments

- In the "Conclusion" section, I suggest giving readers a glimpse of future research on Melanoma detection and classification so that future real-life applications can be based on the results of this work.

Reviewer 2 ·

Basic reporting

The paper presents an AlexNet-based model for classifying melanomas from dermoscopic images.

1.The literature included is relevant, but I suggest referring to all the most recent models in addition to CNNs, such as those based on ViT (visual transformers). Furthermore, I suggest adding references to lines 75 and 161 (that justify why VGG16 with transfer learning outperforms the other techniques). I also suggest a reference to line 190 relating to the MSCD10000 dataset.

2.The text contains some formatting errors; see, for example, line 343 (ref{final}), the space on line 367, and the period before the bracket on lines 116 and 125. I recommend going through the entire paper, looking for similar patterns.

3. The structure of the article conforms to the journal standard. As regards the figures, please insert a more accurate description, in particular in Figure 2 and in Figure 3 (for example, it is not clear for each of the five plots what the combinations of filtering and hair removal techniques are). In Table 1, the acronym BACC is specified, but what balanced accuracy means needs to be clarified. What is meant by Some architecture?

Experimental design

4. The objective of this paper needs to be clarified. The aim is to present a new architecture for classifying melanoma in dermoscopic images. Can you evaluate the impact on the classification of the hair removal process?

5. The details of the parameters relating to the filtering techniques or the hair removal algorithm are not reported in the manuscript, and it would be better to include them for further clarity. Similarly, the Python version and of all libraries are not indicated, as is the computational complexity required for this type of problem.

6. The comparative comparisons provided in Table 4 should be enriched with statistical evaluations to compare the proposed solution's goodness effectively.
Explain the article's objective so that the narrative is consistent from the beginning and highlights the information gap you want to fill.

Validity of the findings

7. The results and conclusions paragraph reports the description of the contents of the tables. However, an integrated evaluation of the consequences of these values ​​is not provided extensively. For example, the Authors declare that the combination of the Wavelet filter with hair removal shows an improvement in accuracy and precision. However, this improvement is minimal and, in any case, not accompanied by information of statistical relevance (for example, what is the effect of the variation of the initialization of network parameters?).

Additional comments

No comment

---

## Round 0.2 · Minor Revisions

Dear authors,

Some minor issues are still required to be addressed. As such, please address all the comments/suggestions provided by the reviewers.

Kind regards,
PCoelho

·

Basic reporting

The authors investigated the use of deep learning for early melanoma detection, focusing on an AlexNet-based CNN model trained on a dataset of 10,605 dermoscopic images. The manuscript was subjected to a complete revision, improving its clarity and structure, as well as the presentation of results and methods.

Experimental design

The experimental design outlined for utilizing deep learning in early melanoma detection, leveraging an AlexNet-based CNN model trained on a dataset of 10,605 dermoscopic images, showcases a thorough and well-considered approach.

Validity of the findings

The validity of the findings in the study on early melanoma detection appears to be well-supported by the substantial dataset. The large and diverse dataset is likely to contribute to the robustness of the model, enhancing its ability to generalize across different instances of melanoma.

Additional comments

A comparison of the model's performance against clinical baselines or human experts could provide valuable insight into its practical utility and areas where it either excels or requires improvement.

Reviewer 2 ·

Basic reporting

The authors have significantly improved the manuscript, addressing most of the concerns raised previously. The manuscript is now clearer and less ambiguous, and the literature references have been improved. The authors have also provided precise references for the dataset. However, I have a few questions. First, there is a discrepancy between the citations for Javed (2021) on line 254 and Javid et al. (2023) on line 260. Are these two citations referring to the same work? Second, for clarity, it would be helpful to label the axes of Figures 3 and 4. Specifically, it is unclear what the units are for the x- and y-axes in Figure 3 (epochs?), and it is unclear what metric is being used to measure performance in Figure 4 (e.g., accuracy, F1 score, etc.).

Experimental design

The research question is now more clearly defined and articulated, and the methods and parameters are now more clearly described and explained. This makes the manuscript easier to understand and follow, and it also strengthens the overall research design.

Validity of the findings

The authors use Figure 4 to illustrate how different filters perform. They suggest a significant difference with the Wavelet filter performing best, particularly when combined with hair removal. However, while the figure shows some variation between filters, it's worth noting that the confidence intervals for the Wavelet filter are also the widest. This could potentially indicate greater variability in its performance. Could you clarify this aspect and provide additional context for interpreting Figure 4?

---

## Round 0.3 · accepted · Accept

Dear authors, we are pleased to verify that you meet the reviewer's valuable feedback to improve your research.

Thank you for considering PeerJ Computer Science and submitting your work.